# No Morphological Integration of Dorsal Profiles in the Araucanian Horse (Colombia)

**DOI:** 10.3390/ani12131731

**Published:** 2022-07-05

**Authors:** Arcesio Salamanca-Carreño, Pere M. Parés-Casanova, David Eduardo Rangel-Pachón, Jannet Bentez-Molano, Oscar Mauricio Vélez-Terranova

**Affiliations:** 1Facultad de Medicina Veterinaria y Zootecnia, Universidad Cooperativa de Colombia, Villavicencio 500001, Colombia; asaca_65@yahoo.es (A.S.-C.); david.rangelp@campusucc.edu.co (D.E.R.-P.); jannet.bentez@campusucc.edu.co (J.B.-M.); 2Acuerdo de Cooperación de Investigación, 25798 Catalonia, Spain; 3Facultad de Ciencias Agropecuarias, Universidad Nacional de Colombia, Palmira 763531, Colombia; ovelez@unal.edu.co

**Keywords:** alloidism, animal morphology, body structure, creole breeds

## Abstract

**Simple Summary:**

Morphological modules are structures that have components which covary strongly, but that in turn are relatively independent of other modules, while morphological integration is understood to mean the coordinated morphological variation of the components of a functional whole. Important traits to describe equine breeds are the profiles of different body regions (alloidism). In this research, it was determined if the division between the cervical, dorsal, and rump profiles has a modular basis as well as a morphological integration. A total of 135 digital photographs were booked, in a lateral view (14 females and 121 geldings; age range: 2–20 years), of adult horses, an equine population typical of the floodplain of Arauca, NE Colombia. From each image, 25 reference points (semi-markers) were obtained at the dorsal level of the neck, back, and croup. The modularity hypothesis of different body profiles based on differentiated regions was tested using the RV coefficient, and a two-block partial least-squares analysis was used to assess the level of morphological integration. The results showed that each alloidic group reflected high integration but low modularity. The absence of the fragmentation of the alloidic assemblages would promote the adaptive capacity of the breed by linking coordinated functional responses to similar selection pressures, for example, field work.

**Abstract:**

The aim of this research was to determine if the division between the cervical, dorsal, and croup profiles (three regions commonly assessed for descriptive profile purposes) has a modular basis as well as a morphological integration. For this, a total of 135 digital photographs were obtained, in a lateral view, of adult horses (14 females and 121 geldings; age range: 2–20 years), of the Araucanian breed, an equine population typical of the flooded savannah of Arauca, NE Colombia. From each image, 25 reference points (semi-landmarks) were obtained at the dorsal level of the neck, back, and croup. The hypothesis of the modularity of different body profiles based on differentiated regions was tested using the *RV* coefficient, and an analysis of two blocks of partial least-squares allowed the evaluation of the level of morphological integration. The results showed that each alloidic group reflected high integration but low modularity. The covariation between the modules was centered mainly on the withers, the loin, and the croup. For the studied profile blocks, no module can be considered. The absence of the fragmentation of the alloidic sets would promote the adaptive capacity of the breed by linking coordinated functional responses to similar selection pressures, e.g., field work. Although the integration between the neck, back, and croup profiles was proven, their modular covariation was low.

## 1. Introduction

Since the beginning of zoo-ethnology, researchers have used morphology to make descriptions and observations of animal groups. These observations are based on differences and similarities among individuals, and have served to make approximations of the relationship between domestic breeds and even ecotypes, being the basis of classical zoo-ethnology elements [1]. However, despite the detailed descriptions, morphology cannot be subject to strong quantitative analyses, and is limited only to structures’ linear measurements. Evidently, this approach has its own limitations, as it does not allow for the real visualization of a form and its possible changes. With the development of new multivariate statistical tools, morphological studies have allowed for the transition from mere descriptions to multivariate quantitative analyses. An interesting example of the application of these methods on the profile of equids (alloidism) is the recent work by the authors of [2].

In geometric morphometrics (GMs), “shape” is defined as any geometric information that remains when the translation, scaling, and rotation effects of an object are removed [3]. In GMs, specific equivalent and homologous points are fixed in the biological structure being studied [4,5]. This is how GMs has made it possible to move from traditional morphometry to focus on the real “shape of structures”. For the location of homologous structures in GMs, two variables are mainly used: outlines and landmarks. Landmarks are anatomical loci that do not alter their topological position relative to other landmarks, providing adequate coverage of a form. Landmarks can be easily and repeatedly located between one organism and another occasion [6]. Sometimes, the analyzed structures are flat or have smooth surfaces on which it is difficult to establish landmarks, or these do not fully cover the section to be analyzed. To solve this limitation, points evenly distributed throughout a structure’s surface are used. These points are known as semi-landmarks [4,7].

A module can be defined as a morphological unit within which exist strong interactions, but at the same time possesses a certain degree of independence from other units [8]. Morphological integration (MI) refers in one way or another to the covariation between modules [9,10,11]. A common method for their study is to analyze the patterns of covariation using a partial least-squares (PLS) regression and the Escoufier *RV* coefficient [12]. Both *RV* and PLS analyses are based on the covariance matrix, although both methods are different and express the covariance in alternative ways [13]. The *RV* coefficient represents the total amount of covariance, scaled by the amount of variation between two sets of variables [14]. Permutation tests are used to evaluate the statistical significance of the *RV* coefficient, i.e., observations in the two sets are randomly permuted to simulate the null hypothesis of complete independence. The PLS regression is a multivariate analysis that uses the singular value decomposition of the covariance matrix between two matrices of partial warps corresponding to two forms (*A* and *B*) that are compared, where the covariation pattern is summarized by two groups of latent vectors (linear combinations), one for each block [14]. These vectors are correlated two to two, so that the first vector extracted from matrix *A* correlates only with the first vector of matrix *B*; similarly, the second vector of *A* is correlated only with the second vector of *B*, etc. This analysis is similar to the canonical correlation method, but the difference is that the vectors are not orthogonal to each other. A PLS analysis performs conformation correlations between structures of the same individual. An *RV* coefficient analysis and a PLS analysis are two complementary methods for exploring covariance patterns, as the former works on the total covariance between two matrices relative to the variance of each matrix, while the latter works on two of the maximal covariance axes [12].

In this article, we describe MI to explore covariation patterns (modularity and integration) between different dorsal profiles of a Colombian creole horse breed using morphometric analyses through the *RV* coefficient and PLS methods.

## 2. Materials and Methods

A sample of Araucanian horses was studied. The Araucanian horse is a creole breed located in the Orinoquia flooded savannah of Arauca, NE Colombia, an area located between 06°02′40″ and 07°06′13″ north latitude and 69°25′54″ and 72°22′23″ west longitude [15,16,17]. It is a large, flooded ecosystem (around 15,000 km^2^) with intensive dry and rainy seasons, where cattle are raised in an extensive way and farmers (“rancheros”) ride horses for their work. Araucanian horses are small—although not elipometrical—and are raised to support the livestock activity. They are bred in a totally extensive way with a diet based on natural grasses [15,16]. The Araucanian horse presents a wide chromatic variety, from basic coats—grey, black, bay, and chestnut—to combined patterns—piebald, appaloosa, and its endurance is notable [15,16,17].

The exclusion criteria of the sampled horses were:(a)Those specimens that contradicted any of the inclusion criteria that identify them as being pure Araucanian.(b)Photographs of low sharpness to distinguish homologous landmarks or images with obvious distortions due to enlargement or deformation.(c)Individuals who presented obvious morphopathological signs.

The final sample was 135 adult animals (14 mares and 121 geldings; age range: 2–20 years) with optimal body conditions from 14 different local farms (“ranchos”). Stallions are not docile, which makes their handling difficult, so these animals were excluded from the present study. An individual digital photograph was obtained in a lateral view on flat ground. Lateral photographs (right or left side; Figure 1) were taken with a Nikon P530 42X optical zoom camera.

A set of 25 reference points (semi-landmarks) in each image were digitized, at the dorsal level of the neck (10 semi-landmarks), back (10 semi-landmarks), and croup (5 semi-landmarks), using the TpsDig v. 1.40 software [18]. These semi-landmarks were then slid on the surface to minimize the thin-plate spline bending energy (TPS).

Once the Cartesian *x*–*y* coordinates of all of the data points were obtained, the shape information was extracted with a full adjustment of Procrustes. A Procrustes overlay is a procedure that removes size, position, and orientation information to standardize and evaluate the joint variation in the whole structure of each specimen based on centroid size. A regression of shape variables on the centroid size (logarithmically transformed) was first performed to calculate the intragroup allometric trajectory through the female–male morphospace. Sex differences were assessed using a Procrustes ANOVA analysis. A principal component analysis (PCA) was then performed at the Procrustes coordinates to explore the morphological variation pattern throughout the sample.

To evaluate a modularity hypothesis, the partition of the homologous landmark configuration into three subsets (cervical, dorsal, and croup) corresponding to the hypothetical modules was specified using the *RV* coefficient. This coefficient can be considered a multivariate analogue of a correlation, and was calculated between the two hypothetical modules and between the sets of alternative partitions, generating a distribution of values and suggesting evidence of block independence, which can be considered as morphological modules. When the values of the *RV* coefficient (between 0 and 1) are <0.5, the covariance of the block is weak. The *RV* coefficient provided by this procedure is the same as the output of the hypothesis modularity tests. When the values of the *RV* coefficient are high, the covariance of the studied blocks is stronger, indicating that the evaluated structure consists of a single integrated unit.

The PLS method was used to study the covariation patterns between two or more variable sets. Its usefulness in GMs lies in the fact that at least one of these two or more sets of variables (or blocks) contains components of the shape. Its objective is to maximize the representation in a few dimensions of the covariance structure between the sets of variables. Views of shape changes were obtained via TPS deformation.

The GMs analysis exclusively considered the variation in profiles. All of the analyses were run using version 1.07a of the MorphoJ software [19], with a significance level of 0.05. Permutation tests (10,000 rounds of randomization) were used to assess statistical significances, i.e., observations were randomly swapped to simulate the null hypothesis of complete independence.

## 3. Results

The shape–size regression was not significant (*p* = 0.119), so an allometric correction was not considered in subsequent analyses. The Procrustes ANOVA for size showed no significant differences between the sexes (F = 0.70; *p* = 0.4059). For the shape, no differences between the sexes were found either (F = 0.68; *p* = 0.795). A MANOVA confirmed these results (Pillai = 0.36; *p* = 0.055). The first two PCs accounted for 65.66% of the total variance (PC1 + PC2 = 35.20% + 30.46%). The space–shape PCA graph of the grouped sample confirmed a total overlap between the sexes (Figure 2). For all of these reasons, in subsequent analyses the grouped data were studied without sex separation.

The distribution of the *RV* coefficients for the three evaluated profiles showed that the a priori hypothesis was found at the left-end side of the distribution curve, although, of the total alternative partitions to which the *RV* coefficient was calculated, some values were lower than the one hypothesized in advance (Table 1). The results present strong evidence to support the low modular organization of the three profiles considered. The PLS values (PLS1 to PLS4) are presented in Table 2, indicating a high covariation between the profiles. The covariation between the modules for PLS1 can be described as centered mainly on the nuchal area (christa nuchae), the withers (regio interscapularis), the loins (regio lumbalis), and the top of the croup (regio sacralis).

## 4. Discussion

A module is defined as a unit within which exist high integration or strong interactions among its components; however, modules are relatively independent of each other. On the other hand, integration refers to the level of cohesion between different structures as the result of interactions between the different biological processes that generate the phenotype under study [13,20]. Morphological integration is usually inferred from the study of the covariation between multiple traits. Therefore, morphological integration usually describes the degree to which one structure is linked to another due to various processes. In any case, both concepts refer to the degree of the covariation between the components of a unit: on the one side, modularity refers to the relative independence in a structure, while morphological integration concerns how these units collide with each other [20,21]. From a morphometric perspective, these interactions would manifest as a strong covariation of the components or parts within a module and a weak covariation between modules.

Among horse breeds, there are marked differences in morphometry and body conditions [22,23], showing differences in dorsal profiles and head sizes in particular [2,24].

Other research has found that the dorsal profile is an indicator of the state of well-being of horses [25,26], as well as in pigs [27]. When the dorsal profile of the back is flat or hollow in the neck and rump, it is an indicator of depressed postures in horses [25]. This is why the evaluation of the dorsal profile has also been used in the assessment of animals’ welfare [28]. As the animals that were studied in this research were in optimal body condition, it was considered that the evaluation of the profiles would be a breed trait, as well as considering that our main purpose was to analyze the covariation (modularity and integration) between profiles on different areas.

In the present study, the *RV* coefficient was used as a general measure of the association strength in profile blocks. Based on the evidence presented here, it is feasible to argue that the components of the profiles are lowly integrated between them, but, through the analysis of PLS, the profiles exhibited a high degree of dependence between them.

What is the nature of these detected interactions? Although the proximate cause of the observed covariation patterns is unknown for us, it probably arises because the dorsal components evolve, develop, and function in a joint and coordinated manner. The absence of the fragmentation of the alloidic sets would promote the adaptive capacity of the breed by linking coordinated functional responses to similar selection pressures, e.g., field work, so a correlated change in the profiles could improve the morphological adaptation as a response to functional requirements. In other words, the correlation could promote alloidic changes in the association between all of the blocks. However, although the phenomenon of the morphological integration between the profiles has been positively proven, it was evidenced that their modular covariation is slight. In our opinion, mechanic demands would be different within each module. For instance, points located more cranially on the cervical profile would be modeled mainly and internally by the M. splenius, while on the withers are present the M. rhomboideus cervici and the cranial part of the M. rhomboideus thoracis. Dorsally to the neck runs the powerful ligamentum nuchae. The hip would be the superficial expression of the powerful M. glutaeus medius. The loin represents the junction of that latter with the M. longissimus lumborum. Highly correlated, integrated traits build up a module (“developmental unit”) since they can share the genetic sources, developmental processes, and mechanic demands acting on them. For the studied profile blocks, no module can be considered. Modularity and integration studies have been carried out in some species of mammals (including humans) [29], but there are few studies on animal breeds [30,31]. This study can be considered as the first research on the morphological integration of dorsal profiles in the Araucanian horse.

This study also showed the usefulness of applying geometric morphometry a to describe morphological shape traits among domestic animals. Geometric morphometric methods offer high flexibility and statistical power, along with well-developed and coherent mathematical foundations. Although the integration between the neck, back, and croup profiles is proven, their modular covariation was low. In the studied case, they ensured a complete and nonredundant characterization in addition to an adequate representation of the dorsal profiles.

## Figures and Tables

**Figure 1 animals-12-01731-f001:**
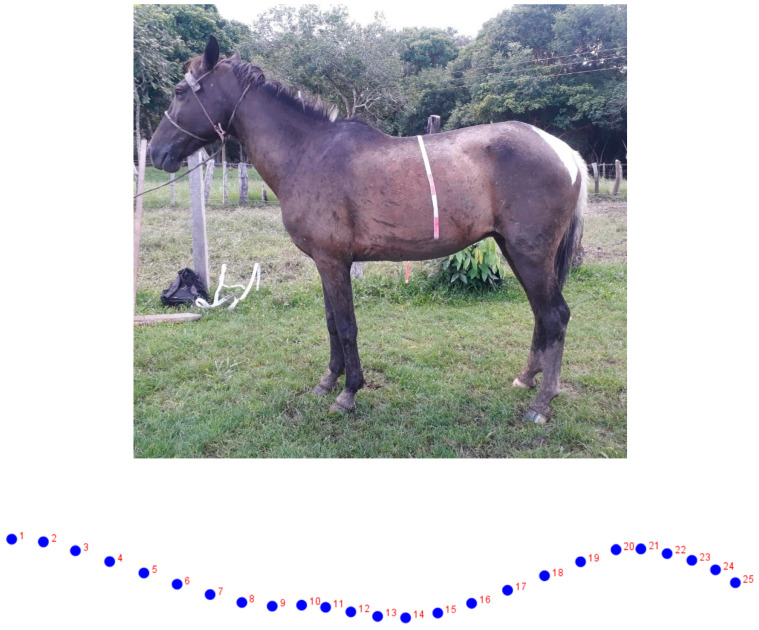
Semi-landmarks used for the study. A set of 25 reference points (semi-landmarks) in each image were digitized at the dorsal level of the neck (10 semi-landmarks), back (10 semi-landmarks), and croup (5 semi-landmarks).

**Figure 2 animals-12-01731-f002:**
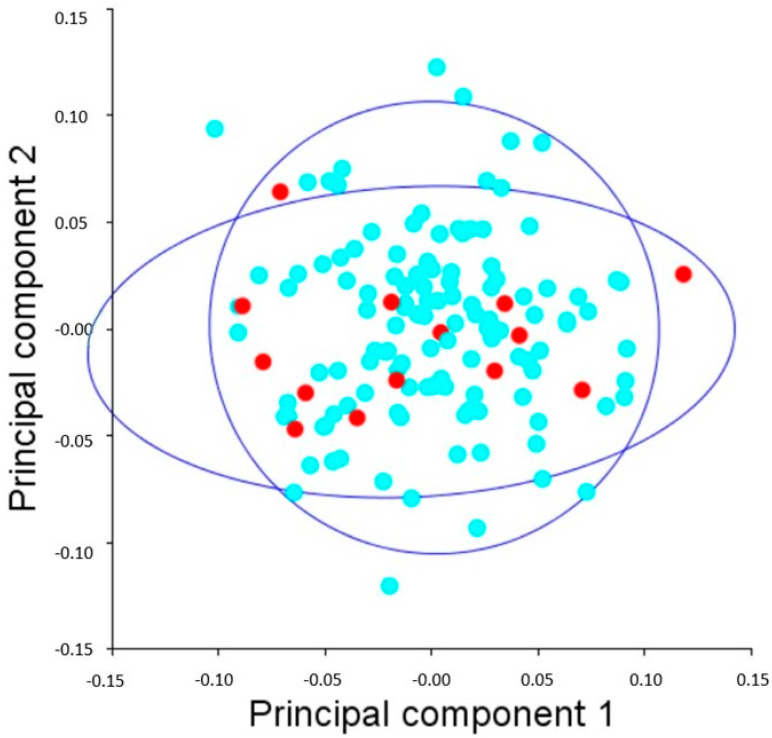
Space–shape PCA for 135 adult animals (14 mares, red points; and 121 geldings, blue points, red and blue points: aggregat) belonging to the Araucanian horse breed. The first two PCs account for 65.6% of the total variance (PC1 + PC2 = 35.20% + 30.46%). It showed a considerable degree of overlap between the sexes.

**Table 1 animals-12-01731-t001:** Results of the modularity analysis by the Escoufier *RV* coefficient (10,000 rounds of randomization) for the different profile units studied.

Semi-Markers	Minimal *RV*	Coefficient *RV*	Number of Partitions with an *RV* Less than or Equal to the a Priori Hypothesis
Neck–back	0.475759	0.475667	7
Neck–croup	0.269920	0.269920	2
Croup–back	0.318913	0.432302	371

**Table 2 animals-12-01731-t002:** Partial least-squares (PLS) values (% total variation) for the different studied profile units.

Semi-Markers	PLS1	PLS2	PLS3	PLS4
Neck–back	87.871	4.729	3.398	1.549
Neck–croup	80.483	9.666	7.643	1.059
Croup–back	91.742	3.727	2.153	1.318

## Data Availability

The data are available upon reasonable request to the second author.

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
