# Peer review of "No Morphological Integration of Dorsal Profiles in the Araucanian Horse (Colombia)"

_animals, 2022, doi:10.3390/ani12131731_

Round 1
Reviewer 1 Report
This study is interesting and inspiring material for research workers specializing in animal morphology (alloidism), however has no practical aspects, especially what concerns diagnosis. I would recommend to compare integration between neck, back and croup profiles in different equine breeds, ie., Araucanian horse vs throughbred horse. Such comparison can be extremely interesting. Other criticism: small population for cohort study- only 136 horses (in similar human studies authors usually deal with a couple thousands of participants) , very wide age range - horse 2 years old is completely unlike the horse 20 years old, disproportion between female and male- 14 vs 121. In conclusion, there are errors in methodology. Design of the experiment could be also improved.
Author Response
Respuestas del primer revisor
Comentario
Este estudio es un material interesante e inspirador para los investigadores especializados en morfología animal (aloidismo), sin embargo, no tiene aspectos prácticos, especialmente en lo que respeta al diagnóstico. Recomendaría comparar la integración entre los perfiles de cuello, espalda y grupa en diferentes razas equinas, es decir, caballo araucano vs caballo purasangre. Tal comparación puede ser extremadamente interesante.Otras críticas: población pequeña similar para el estudio de cohorte: solo 136 caballos (en estudios humanos, los autores generalmente tratan con un par de millas de participantes), rango de edad muy amplio: el caballo de 2 años es completamente diferente al caballo de 20 años, desproporción entre hembras y masculino- 14 vs 121. En conclusión, hay errores en la metodología. El diseño del experimento también podría mejorarse.
Respuesta
La diferencia entre la población de yeguas y machos se debe a que los caballos castrados son más dóciles de manejar y están acostumbrados a trabajar, manejar ganado y facilitar la toma de datos. Los sementales son más inquietos, lo que dificulta el manejo para la recolección de datos. Por edad, se puede considerar que los animales mayores de 2 años presentan un perfil definitivo. Para evitar sesgos por mala condición corporal (especialmente en los perfiles dorsales), solo se muestrearon animales en buen estado.
La comparación con otras razas no era nuestro propósito. Además, hay una falta total de investigaciones similares. De todos modos, nuestro objetivo -y la principal novedad de esta investigación- era analizar la covariación (modularidad e integración) entre tres perfiles corporales diferentes, utilizando técnicas de morfometría geométrica.
Todos estos puntos han sido introducidos en el manuscrito. Se añaden siete citas bibliográficas en la discusión.
Inglés corregido

Reviewer 2 Report
It would be interesting to conduct a comprehensive analysis of the modules of the entire physique of horses of this breed and compare the results obtained with data on Arab and Thoroughbred horses.
Author Response
Respuestas del segundo revisor
Comentario
Sería interesante realizar un análisis exhaustivo de los módulos de todo el físico de los caballos de esta raza y comparar los resultados obtenidos con los datos de caballos árabes y pura sangre.
Respuesta
Se añaden siete citas bibliográficas en la discusión, aunque, como hemos dicho anteriormente, no existen trabajos similares sobre este tema (y, como se ha dicho, tampoco era el objeto de nuestro estudio).
Inglés corregido

Reviewer 3 Report
Dear Authors, please correct the description of the tables!

Author Response
Respuestas del tercer revisor
Comentario
Estimados autores, corrijan la descripción de las tablas.
Respuesta
corregido